# Conserved transcriptomic profile between mouse and human colitis allows unsupervised patient stratification

Paulo Czarnewski[1], Sara M. Parigi[1], Chiara Sorini [1], Oscar E. Diaz [1], Srustidhar Das[1], Nicola Gagliani[1,2,3] & Eduardo J. Villablanca [1,3]

Clinical manifestations and response to therapies in ulcerative colitis (UC) are hetero-geneous, yet patient classification criteria for tailored therapies are currently lacking. Here, we present an unsupervised molecular classification of UC patients, concordant with response to therapy in independent retrospective cohorts. We show that classical clustering of UC patient tissue transcriptomic data sets does not identify clinically relevant profiles, likely due to associated covariates. To overcome this, we compare cross-sectional human data sets with a newly generated longitudinal transcriptome profile of murine DSS-induced colitis. We show that the majority of colitis risk-associated gene expression peaks during the inflammatory rather than the recovery phase. Moreover, we achieve UC patient clustering into two distinct transcriptomic profiles, differing in neutrophil-related gene activation. Notably, 87% of patients in UC1 cluster are unresponsive to two most widely used biological therapies. These results demonstrate that cross-species comparison enables stratification of patients undistinguishable by other molecular approaches.

[1] Immunology and Allergy Unit, Department of Medicine, Solna, Karolinska Institute and University Hospital, 17176 Stockholm, Sweden. [2] Department of Medicine and Department of General, Visceral and Thoracic Surgery, University Medical Center Hamburg-Eppendorf, 20246 Hamburg, Germany. [3] These authors jointly supervised this work: Nicola Gagliani, Eduardo J. Villablanca. Correspondence and requests for materials should be addressed to E.J.V. (email: eduardo.villablanca@ki.se)

Ulcerative colitis (UC) is a type of inflammatory bowel disease (IBD) that is mostly restricted to the colon and is characterized by changes in the mucosal architecture, epithelial function, increase in immune cell infiltration, and an elevated concentration of inflammatory cytokines. Symptoms include diarrhea, abdominal pain, rectal bleeding, lack of appetite, and fatigue, all of which significantly affect the patient's quality of life. UC is a heterogeneous disease, presenting diverse macroscopic features, symptoms, grades of inflammation, and colonic affected areas[1,2].

Although there is no definitive cure for UC, there are biological therapies available, which target the inflammatory response during UC by means of inhibiting proinflammatory cytokines or by blocking immune cell migration[3]. Among these, the most frequently used biological therapies in UC patients block tumor necrosis factor (TNF) with anti-TNF antibodies (such as infliximab, IFX)[4] or leukocyte migration (such as vedolizumab, VDZ)[5,6]. However, about 35[4,6] and 50%[5,6] of patients poorly achieve clinical response to IFX and VDZ, respectively. Patients that do not respond develop adverse effects, most notably increased risk of infections, thus requiring continuous medical monitoring and ultimate surgical intervention[7,8].

In an attempt to identify genes/pathways as a potential novel therapeutic target, genome-wide association studies (GWAS) have identified more than 200 polymorphisms associated with a higher susceptibility to IBD[9,10]. However, the function and temporal expression of IBD-risk genes during experimental colitis are yet to be elucidated[9,10].

Furthermore, while there is an obvious clinical heterogeneity among UC patients, as seen, for example, by the location affected (i.e., distal colitis, left-sided and pancolitis, and responder and nonresponder) and the extent of the severity, initial treatment for these patient profiles is identical and modified only if the patients have not responded[6,8]. Biomarkers that could distinguish the different entities of the UC spectrum are currently lacking and they are required in order to achieve the highly needed stratification of UC patients into molecularly functional transcriptomic profiles[8,11]. Moreover, an unbiased stratification of UC subtypes has not been reported at the molecular and functional levels, to the best of our knowledge. Here, using transcriptomic data from a well-characterized experimental model of colitis, we identify conserved genes between mouse and UC patients. As a result, we offer insights into IBD-risk gene kinetics and to molecularly stratify UC patients in an unsupervised manner.

## Results

### Human UC is highly variable at the transcriptome level

In order to stratify UC patients into molecular profiles, we combined four publicly available human UC cohort data sets ($n = 102$ patients), in which transcriptomic microarrays of the total colonic biopsies were performed[12–15] (Table 1 and Supplementary Fig. 1). We ranked genes using the top 100 most variable genes and further tested whether transcriptomic profiles exist (Fig. 1a). Analysis by visual assessment of cluster tendency (VAT)[16] indicated that biopsies presented high inter-sample dissimilarities (Fig. 1b), suggesting a poor overall tendency to form consistent clusters. Dimensionality reduction analysis by t-SNE using the top highly variable genes also indicates the formation of a single group with no apparent subdivisions (Fig. 1c). Then, we further statistically tested whether multi-cluster substructures were present in the data set, since most clustering algorithms define transcriptomic profiles even on random noise[17–19]. However, bootstrapping analysis using the Hartigan's Dip test[19,20] presented a low cluster substructure trend ($p > 0.9$), regardless of the gene-ranking metrics used (Fig. 1d). Independently of the

**Table 1 Publicly available human data sets used in this paper**

| Data set ID | Total | Responders | Nonresponders | Ref. |
|---|---|---|---|---|
| Infliximab: | | | | |
| GSE12251 | 23 | 11 | 12 | 13 |
| GSE73661 | 23 | 15 | 8 | 15 |
| GSE23597 | 32 | 7 | 25 | 14 |
| GSE16879 | 24 | 16 | 8 | 12 |
| Sum | 102 | 49 | 53 | |
| Vedolizumab: | | | | |
| GSE73661 | 37 | 23 | 14 | 15 |
| Pediatric UC: | | | | |
| GSE109142 | 206 | 105 | 101 | 33 |

Data sets used for the classification of ulcerative colitis molecular profiles. Only the number of patients used for analysis are shown (inflamed mucosa before receiving any therapy)

clustering tendency results, we forced patient subdivision using hierarchical clustering and tested for cluster stability using bootstrapping[17,18,21]. In line with previous results, formed clusters were highly unstable using the list of highly variable genes (AU ≈ 0%) (Fig. 1e). These results indicate that without prior knowledge of patient subdivision, standard gene-ranking strategies do not allow clustering of UC patients into consistent molecularly distinct profiles.

### Time-series analysis during colon inflammation and repair

One cause of such inter-patient variability can be attributed to the sampling procedure, which contributes largely to the total data variance and masks real biological differences[22,23]. To overcome the total data variance, we sought to identify the genes that contribute to inflammation in an independent and unsupervised manner. To this end, we focused the analysis on a list of evolutionarily conserved genes that best discriminate the nuances of inflammation in a well-characterized colitis mouse model[24].

To identify these evolutionarily conserved genes, we first elucidated through an unbiased manner which genes and pathways are differentially regulated during mouse colonic inflammation, followed by a tissue regeneration phase. In particular, we took advantage of the widely used dextran sodium sulfate (DSS)-induced model of colitis. This model is one of the few characterized by a phase of damage, followed by a phase of regeneration. Therefore, this model gave the possibility to identify also sets of genes essential in the regeneration phase, a key step toward the resolution of the inflammation. In short, mice were exposed to DSS in the drinking water for 7 days, and then allowed to recover for the following 7 days. During this period, we collected colonic tissue samples every second day to then be analyzed by RNA sequencing (RNA-seq), histology, and flow cytometry (Fig. 2a and Supplementary Fig. 2). First, we confirmed that 7 days of DSS exposure resulted in continuous body weight loss and acute disease severity, until day 10 to then initiate the recovery phase (Supplementary Fig. 2a–b). Histological analysis confirmed epithelial damage, such as desquamation of the epithelial layer on day 6 (Supplementary Fig. 2c), while labeling proliferating cells within crypts (Ki67 staining) indicated a disrupted crypt architecture by day 6 and restoration by day 14 (Supplementary Fig. 2c). Loss of the epithelial cells ($CD45^{neg}Ep\text{-}CAM^+$) by day 7–10 and restoration by day 14 was further confirmed by flow cytometry (Supplementary Fig. 2d). To test whether the epithelial barrier integrity was restored by day 14, we gavaged FITC dextran and measured its concentration in the serum. We detected higher FITC-dextran concentrations on day 7, which indicate barrier disruption, whereas basal levels were detected by day 14, indicating restoration of the barrier integrity

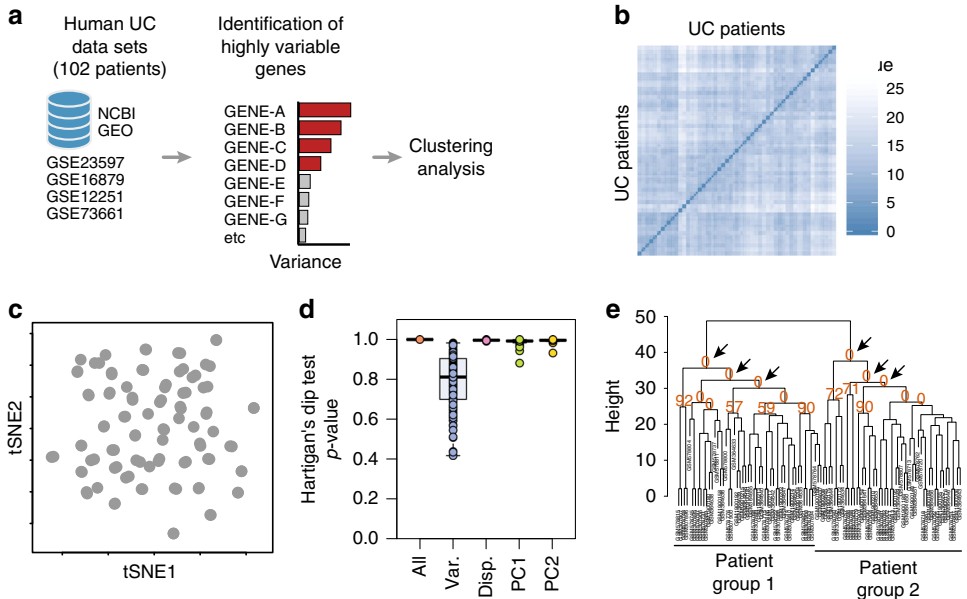

**Fig. 1** Human ulcerative colitis transcriptional signature does not cluster patients into molecular profiles. **a** Schematic representation of the strategy used for patient group identification, in which four publicly available data sets were combined. Gene ranking was done using the most variable genes in the human data set, which were used for clustering analysis. **b** Sample dissimilarity heatmaps for visual analysis of clustering tendency (VAT), comparing the human data set using the top 100 variable genes. **c** t-SNE plot using the top 100 variable genes in the human data set. Each point represents a patient sample. **d** Hartigan's Dip test for clustering tendency using all genes in the data set, the top 100 variable genes, the top 100 highly dispersed genes, or the top 100 leading genes in the principal components. **e** Bootstrapping analysis of hierarchical clustering, comparing the human data set using the top 100 variable genes in the human data set. Numbers in orange indicate the approximately unbiased (AU) p-value, shown as a percentage. AU closer to zero indicates a cluster with low stability. Boxplot represents the median (center line, second quantile), first, and third quantiles (box) and whiskers extending 1.5 times the interquartile range (IQR)

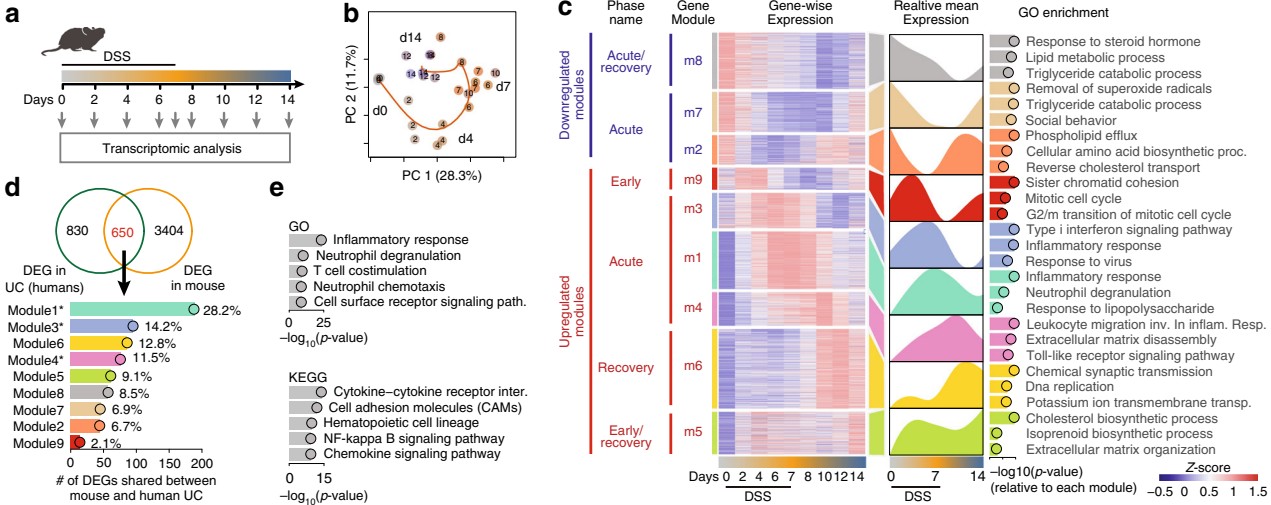

**Fig. 2** Unbiased characterization of the DSS colitis reveals conserved inflammatory signature between mice and humans. **a** Schematic illustration of the experimental design. Mice received DSS in drinking water for 7 days, after which the treatment was replaced with water. Samples were collected at indicated time points. **b** PCA on differentially expressed gene counts. Samples were color-coded according to their respective day of collection (from gray to orange to blue). The percentage of variance explained by the respective principal component is indicated in parenthesis. **c** Clustered heatmap of all differentially expressed genes (left). The mean expression of each gene module is shown (right). Functional annotation of genes in each cluster was done based on Gene Ontology (GO) enrichment. Only the top three enriched processes are shown, sorted by p-value. **d** Venn diagram comparing the list of DEGs in treatment-naive UC and the DEGs identified in mouse DSS colitis (upper). Among the 650 genes shared among those lists (in red), the number and percentage of genes were found in each module identified in our mouse data set (lower). Modules highlighted in bold are the ones enriched for inflammatory terms in (**c**). **e** GO and KEGG enrichment analysis out of the 650 shared genes identified in (**d**), sorted by p-value

(Supplementary Fig. 2e). Thus, on the basis of this characterization, we will refer to d6–d10 and d12–d14 as acute phase and recovery phase, respectively.

Next, we performed a RNA-seq analysis from colonic samples throughout the experiment and computed differentially expressed genes (DEGs), taking the complete kinetics of expression into consideration for *p*-value estimation using EdgeR[25] (see the "Methods" section). A detailed list of all genes found differentially expressed is available for further exploration (Supplementary Data set 1). Principal component analysis (PCA) on DEGs revealed that samples displayed a sequential temporal path in PCA space, starting on day 0, passing through day 7 (acute), and ultimately reaching day 14 (recovery) (Fig. 2b). Of note, samples from day 14 did not reach the same gene expression profile compared with day 0, suggesting that complete molecular restoration was not reached by day 14. We observed that over 70% of the variance among the differentially expressed transcripts is retained in the first five principal components (PCs) (Supplementary Fig. 3a), and that each principal component corresponds to a unique expression kinetics through the time course of DSS colitis (Supplementary Fig. 3b). For instance, the variance explained by PC1 peaked at the acute phase and returned to almost normal levels on day 14 (recovery), capturing most of the variance related to inflammatory genes that peaked from days 7 to 10, such as *Ly6g*, *Reg3b*, *Reg3g*, *S100a8*, *S100a9*, *Mmp3*, *Mmp8*, and *Mmp10* (Supplementary Fig. 3b and c). On the other hand, the variance explained by PC2 peaked on day 4 during DSS administration, to return close to normal by day 7, thus capturing most of the variance related to genes expressed during initiation of inflammation, such as *Mcpt1*, *Mcpt2*, *Mmp3*, *Mmp10*, *Il11*, *Scnn1g*, and *Best2* (Supplementary Fig. 3b and c). These results indicate that several of the genes modulated between days 4 and 10 are related to inflammation and together contribute the most to the variance in the data set.

By using hierarchical clustering on the spline-smoothed gene expression of DEGs, we were able to classify the gene expression into nine modules (Fig. 2c). For further exploration, expression values for all genes in each module are available (Supplementary Data set 1). Three gene modules (m2, m7, and m8) were downregulated during the acute and recovery phases of DSS-induced inflammation, with the lowest peak on days 6, 10, and 12, respectively. GO and KEGG enrichment analysis suggest that these modules represent genes mainly involved in epithelial cell functions, such as PPAR signaling (*Acsl1*, *Fabp1*), small-molecule metabolism (*Sult1a1*, *Sult1b1*), and fat digestion and absorption (*Paqr8*, *Clps*, and *Pla2g3*) (Fig. 2c and Supplementary Fig. 4a).

On the other hand, six modules (m9, m3, m1, m4, m6, and m5) were upregulated over the early, acute, and recovery phases of DSS-induced inflammation, peaking on days 2, 6, 7, 10, 12, and 14, respectively. Among those, processes such as cytokine signaling (*Il11*, *Il12b*, *Il6*, and *Il1b*), leukocyte migration (*Sell*, *Ccr1*, *Ccr2*, *Cxcl2*, and *Cxcr3*), neutrophil degranulation (*Ly6g*, *Itgam*, *Itgax*, and *Cd300a*), matrix remodeling (*Mmp3*, *Mmp7*, and *Mmp10*), response to lipopolysaccharide (*Saa3*, *Nox2*), as well as several inflammatory signaling pathways (*Stat3*, *Jak3*, *Nfkbia*, *Smad4*, and *Birc3*) were enriched, suggesting the interplay of several immune cells and pathways as a cause/trigger of inflammation, especially during the acute phase (Fig. 2c and Supplementary Fig. 4b). Moreover, modules m9 and m5 presented two degrees of bimodal expression pattern, peaking at days 2–4 (early phase), with a slight downregulation between days 7 and 10 and a second peak on days 12–14 (recovery phase). Genes in those modules were associated mainly with cell cycle (*Ttk*, *Cdc7*, *Cdc20*, *Cdc25c*, *Ccna2*, *Ccnb1*, and *Ccnb2*) and cholesterol biosynthetic pathways (*Acat2*, *Sqle*,

*Mvd*, and *Hmgcs1*), respectively (Fig. 2c and Supplementary Fig. 4b). Many other genes and GO/KEGG pathways not shown here are fully accessible for exploration of individual genes and their clusters (Supplementary Data sets 1, 2 and 3). Taken together, time-series transcriptomic characterization of mouse colonic inflammation identifies distinct gene expression kinetics associated with epithelial and immune cell-related pathways during the course of colitis.

**Inflammatory pathways conserved between mice and humans.** Having characterized genes and pathways that are associated with intestinal inflammation and tissue repair during experimental colitis, we investigated whether such pathways are conserved in humans. To this end, we compared the list of DEGs from the mouse experimental colitis with the recently published list of DEGs found in newly diagnosed treatment-naive ulcerative colitis patients[26]. This is a cohort containing human RNA-seq data, where they report DEGs between UC patients versus healthy controls. We found that among the 4045 mouse DEGs, 650 genes were also found among the list of DEG obtained comparing UC patients versus healthy controls (Fig. 2d and Supplementary Data set 4). Out of the 650 genes shared between mouse and humans, 53.9% were identified in the inflammatory modules m1 (28.2%), m3 (14.2%), and m4 (11.5%) (Fig. 2d). This suggests that acute inflammatory genes in m1, m3, and m4 are conserved between DSS-induced colitis and UC. GO and KEGG enrichment analysis revealed that those 650 genes were enriched for inflammatory pathways related to neutrophil degranulation and chemotaxis, as well as cytokine and inflammatory signaling pathways (Fig. 2e and Supplementary Data set 5). These results showed that most of the genes/pathways conserved between experimental mouse colitis and human UC are associated with inflammatory responses.

**Temporal classification of IBD-risk genes.** To understand the temporal expression of the genes associated with the identified IBD polymorphisms (candidate IBD-risk genes)[9], we checked the expression of genes associated with UC or CD identified by single-variant fine-mapping resolution[10] into the list of DEGs from the mouse data set. Out of the 233 reported candidate IBD-risk genes, 40 genes presented very low or undetectable counts in the mouse data set (i.e., *IL23R*, *SULT1A2*, *ERAP2*, and *MUC19*), 118 were detected but did not have their expression altered through the development of inflammation (i.e., *TNFRSF14*, *ATG16L1*, *GPR35*, and *TNFSF8*), and 75 were found among the DEGs in our mouse data set (Supplementary Fig. 5a and Supplementary Data set 6). Among these, many IBD-risk genes with already-known functions during mouse colitis were found (e.g., *IFNG*, *GPR65*, *ITGAL*, *CCL7*, *STAT3*, *FUT1*, *CD40*, *SULT1A1*, *MUC1*, *CARD9*, *IL12B*, *IRF1*, and *CD5*), being specifically present in gene modules related to inflammation m1, m3, and m4. Moreover, 26 genes of the 75 IBD-risk genes found in our data set are shared between UC and CD (i.e., *CARD9*, *SULT1A1*, *STAT3*, *GPR65*, and *IL12B*), while 10 and 39 were restricted to UC or CD, respectively (Supplementary Fig. 5b and Supplementary Data set 7). In order to provide temporal information regarding the expression of IBD-risk genes during inflammation and repair, we utilized the mouse transcriptional landscape to map at which time point homolog IBD-risk genes were up- or downregulated. Out of the 75 genes shared between mouse DEGs and IBD-risk genes, 45 (60%) were mapped to modules m1, m3 and m4, which represent the acute phase of inflammation (Supplementary Fig. 5c and Supplementary Data set 7). Among them, we found *Card9, Ifng, Il12b, Stat3, Stat4,* and *Cd40*, which have been reported to exert functions during the acute phase of intestinal inflammation[27–32].

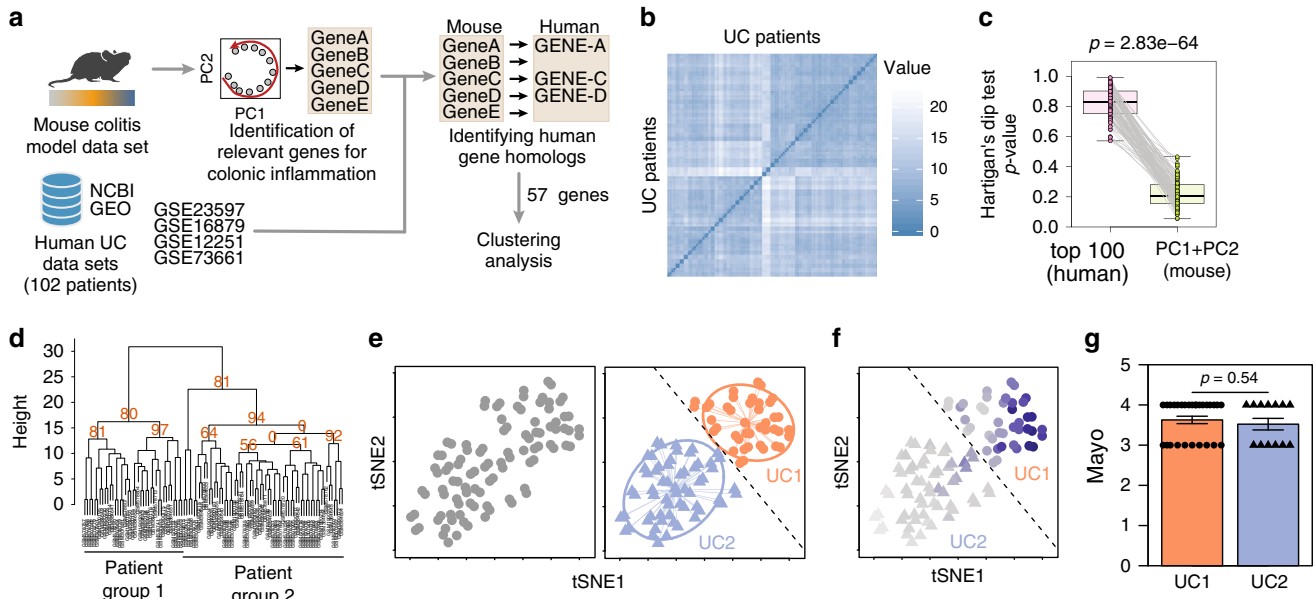

**Fig. 3** Conserved inflammatory gene signature distinguishes two UC transcriptomic profiles. **a** Schematic representation of the strategy used for patient group identification. Four publicly available data sets were combined. Gene ranking was done using the most variable genes-identified mouse data set that had a homolog in humans. **b** Sample dissimilarity heatmaps for visual analysis of clustering tendency (VAT), comparing the human data set using the top mouse gene homologs. **c** Hartigan's Dip test for clustering tendency comparing the analysis using the top 100 variable genes and the top mouse gene homologs. **d** Bootstrapping analysis of hierarchical clustering, comparing the human data set using the top mouse gene homologs. Numbers in orange indicate the approximately unbiased (AU) p-value, shown as percentage. AU closer to zero indicates a cluster with low stability. **e** t-SNE plot using the top variable genes identified from the mouse data set. Each point represents a patient sample. t-SNE plot showing the separation of two patient profiles (left). Unsupervised hierarchical agglomerative clustering was used to automatically define patient subdivision (center). Dashed line delimits UC1 (triangle) and UC2 (circles) patients. **f** Average expression of mouse homolog genes used to subdivide patients, where the dark blue color indicates a higher average expression. Dashed line delimits UC1 (triangle) and UC2 (circles) patients. **g** Assessment of Mayo clinical subscore in patients from UC1 and UC2. Mann–Whitney test was used for comparison. Boxplot represent the median (center line, second quantile), first, and third quantiles (box) and whiskers extending 1.5 times the interquartile range (IQR). Error bars represent the standard deviation

By contrast, *Fut1, Sult1a1, Hes5*, and *Tnfsf15* were mapped to modules m8, m7, and m2, which are downregulated during acute inflammation, while *Rasip, Ntn5*, and *Rtel1* matched with module 6, which is associated with genes that are upregulated during the recovery phase after acute inflammation (Supplementary Fig. 5c). These data thus provided temporal information on when IBD-risk genes are differentially expressed during damage and tissue repair, providing useful insights into their potential roles during inflammation and recovery.

**Conserved genes distinguish two UC transcriptomic profiles.** Having identified genes that contribute to inflammatory pathways that are conserved between mice and humans, we next used those genes to assess whether UC patients can be subdivided into distinct transcriptomic profiles (Table 1, Fig. 3a). To this end, we selected the top 100 leading genes in PC1 and PC2 from the mouse colitis data set and identified the respective human homologs (Fig. 3a). We found that 57 genes were shared between mice and humans. Of these, only 17 genes were found among the 100 most variable genes of the human data set (Supplementary Fig. 6), which might explain why patient classification using highly variable genes was not possible.

Therefore, we performed an unsupervised analysis of the human data set, using 57 homolog genes (Fig. 3a). Of note, VAT analysis using these 57 homolog genes indicated the distinction into two major patient transcriptomic profiles (Fig. 3b), which also resulted in reduced Hartigan's unimodality test ($p < 0.001$, Fig. 3c). This indicates that by using mouse most variable genes as opposed to the sole top human variable ones, it is possible to

obtain a higher clustering tendency of the UC patient data. To test whether using the mouse homologs also impacted on cluster stability, we performed a bootstrapping analysis. This time, clustering using the top mouse homolog genes resulted in clusters with a higher stability (AU ≈ 80%) (Fig. 3d), compared with using the top human highly variable genes (AU ≈ 0%) (Fig. 1e). Hierarchical agglomerative clustering using the mouse homolog genes thus defined two UC transcriptomic profiles, namely UC1 and UC2, comprising 60 and 42 patients, respectively (Fig. 3e). The UC1 transcriptomic profile is defined as patients presenting the higher average expression of the inflammatory genes compared with UC2 (Fig. 3f). We also observed that neither UC1 nor UC2 profiles were discriminated by the overall macroscopic tissue disease severity (Fig. 3g), suggesting that although these two UC profiles are indistinguishable based on the histological Mayo score, they are transcriptionally distinct.

**Key genes allow the distinction between UC1 and UC2 profiles.** In order to characterize UC1 and UC2 beyond conserved genes, we performed differential expression analysis using all genes present in the human data set. We were able to identify 205 highly differentially expressed genes, among which 187 were upregulated in UC1 and 18 were upregulated in UC2 (Fig. 4a). Detailed tables with information on all DEGs comparing UC1 and UC2 are available for exploration (Supplementary Data set 8 and Supplementary Fig. 7a). Among those, cytokines (*TNF, IL11*), enzymes (*NOX1, MMP3*, and *CYP26B1*), calcium-binding proteins (*S100A8, S100A9*), chemokines (*TREM1, CXCL8*), and other

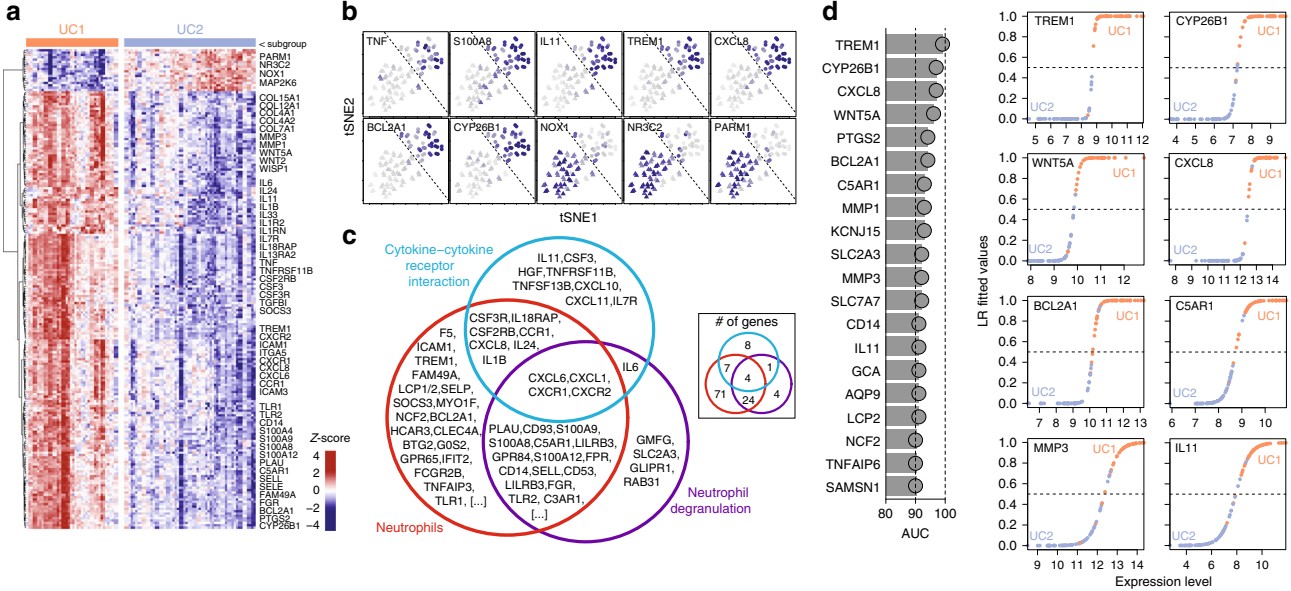

**Fig. 4** UC1 transcriptomic profile is enriched for the inflammatory signature. **a** Heatmap of DEGs between UC1 and UC2 patients, including all genes in the human data set. Only the selected genes are shown, grouped by functional categories and respective to the expression level. **b** t-SNE overlay of the expression level of selected DEGs between UC1 and UC2, showing inter-patient variation. **c** Venn diagram of the top GO, KEGG, and cell-enriched terms identified from the DEGs between UC1 and UC2. **d** Top 20 genes ranked by area under the curve (AUC) for specificity and sensitivity to distinguish UC1 from UC2, among the list of DEGs (left). Classification was carried out using logistic regression. The fitted values of prediction are shown for selected genes (right)

proteins related to the inflammatory response (*NR3C2*, *BCL2A1*, *PARM1*, and *TNFSF13B*) were clearly able to discriminate UC1 from UC2 (Fig. 4b and Supplementary Fig. 7). Enrichment analysis for cell types, GO, and KEGG pathways revealed that genes highly expressed in UC1 (187) were associated with terms related to neutrophil, neutrophil degranulation, and cytokine–cytokine receptor interaction, respectively (Supplementary Fig. 7b). Venn diagram of the top enriched terms revealed that many overlapping genes are shared among these pathways (Fig. 4c), suggesting that UC1 patients present a distinct transcriptional signature enriched in neutrophil activity and cytokine signaling compared with UC2 patients.

We trained a logistic regression classifier, using each of the DEGs between UC1 and UC2 to identify key genes that could be further used in the clinics for distinction of UC1 and UC2. Genes were tested and scored individually, using the area under the curve (AUC) as a combined measure of sensitivity and specificity (Fig. 4d). We observed that genes such as *TREM1* (AUC = 99%), *CYP26B1* (AUC = 97%), and *CXCL8* (AUC = 97%) were among the top markers to distinguish UC1 from UC2. Other genes such as *WNT5*, *BCL2A1*, *C5AR1*, *MMP1*, *MMP3*, and *IL11* also presented AUC scores above 90% and also represented good candidates for UC1 and UC2 distinction in clinical practice.

**UC1 and UC2 respond differently to biological therapies.** While we stratified UC patients into two molecularly distinct profiles, it was unclear whether UC1 and UC2 show different treatment responses to biological therapies. To address this, we used the patient-specific treatment response obtained 4–8 weeks after the biopsy was taken and treatment with IFX started (Table 1). Interestingly, we observed that on average, 70% of the patients belonging to the UC2 transcriptomic profile responded to infliximab therapy (Fig. 5a), in contrast to <10% of the patients classified as UC1, regardless of the data set analyzed (Fig. 5a).

To extend the applicability of our findings, we made use of another set of UC patients receiving vedolizumab and repeated the same procedure as before (Table 1). Transcriptomic data from UC patients were analyzed using the most relevant genes identified in our mouse colitis model and then clustered as described above to reveal UC1 and UC2. Between them, UC1 presented a higher expression of the conserved inflammatory genes (Fig. 5b). We observed that about 60% of the patients belonging to the UC2 transcriptomic profile responded to VDZ, in comparison with about 13% of the patients belonging to the UC1 transcriptomic profile (Fig. 5c). Taken together, the data indicate that patients belonging to the UC2 transcriptomic profile, which presents a higher percentage of response, respond to either IFX or VDZ treatment. Importantly, our approach actually allows a more accurate identification of those patients with UC1, in which 87% of the patients are refractory to both IFX and VDZ.

Next, we tested the robustness of our cross-species unbiased approach to classify UC1 and UC2 patients, using a data set in which a distinct IBD onset (pediatric) and methodology to obtain gene expression profiles (RNA-seq) was used[33]. Hence, using RNA-seq data from 206 pediatric UC patients (GSE109142)[33] (Table 1), we were able to identify two major transcriptomic profiles, in which the enrichment of inflammatory genes distinguished the UC1 from the UC2 transcriptomic profile (Fig. 5d). Differentially expressed genes between UC1 and UC2 identified using microarray data (e.g., TREM1) were also observed in analysis from RNA-seq data (Fig. 5e). Moreover, we observed that 107 of the DEGs were shared between microarray and RNA-seq data sets (Supplementary Fig. 8a and Supplementary Data set 9), among which the key genes used to differentiate UC1 from UC2 were found (see Fig. 4). We further compared the p-values and the expression trends of those 107 DEGs found in both data sets. We observed that those genes are highly significant, regardless of the data set used (Supplementary Fig. 8b), and that 100% of them have the exact same expression

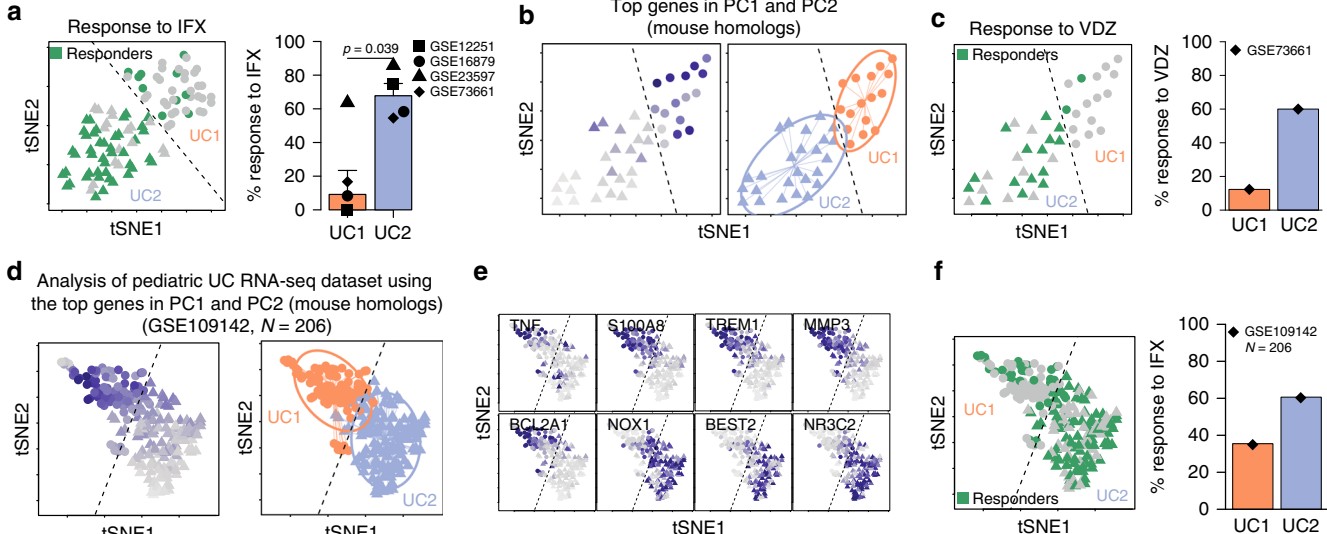

**Fig. 5** UC1 and UC2 differ on their response to IFX and VDZ therapy. **a** Individual patient response to IFX therapy in each group and the percentage of patients responding to IFX in each cohort. **b** t-SNE plot using the top variable genes identified from the mouse data set. Average expression of mouse homolog genes used to subdivide patients (left), where dark blue color indicates a higher average expression. Unsupervised hierarchical agglomerative clustering was used to automatically define patient subdivision (right). Dashed line delimits UC1 (triangle) and UC2 (circles) patients. **c** Individual patient response to VDZ therapy in each group and the percentage of patients responding to VDZ in the cohort (right). **d** t-SNE for data set GSE109 containing 206 pediatric UC patients plot, using the top variable genes identified from the mouse data set. Each point represents a patient sample. The average expression of mouse homolog genes used (left) to subdivide patients into UC1 or UC2 (right). Unsupervised hierarchical agglomerative clustering was used to automatically define patient subdivision (right). Dashed line delimits UC1 (triangle) and UC2 (circles) patients. **e** t-SNE overlay of the expression level of selected DEGs between UC1 and UC2, showing inter-patient variation in the pediatric UC cohort. **f** Individual patient response to IFX therapy in each group and the percentage of patients responding to IFX in each cohort

trend, i.e., meaning that genes found upregulated in the microarray data set were also found upregulated in the RNA-seq data set, and vice versa (Supplementary Fig. 8c). Thus, molecular stratification into UC1 and UC2 can be achieved, regardless of the methodology used to obtain gene expression profiles. Similarly to the adult onset, we did not observe any differences in sex distribution, age, histological score, or calprotectin levels between UC1 and UC2 from the pediatric onset (Supplementary Fig. 9a). However, pediatric UC1–UC2 classification was found to be associated to the overall Mayo and Pucai scorings (Supplementary Fig. 9b), suggesting that these general disease scoring methods could already be used as tools to help discern UC1 and UC2 profiles in pediatric IBD. Thus, our results indicate that the UC1 and UC2 identification is conserved between adults and pediatric UC, with many key marker genes, regardless of patient age and methodology used for RNA profiling. Finally, in line with previous results on adult UC patients, 60% of patients classified as UC2 clinically responded to IFX, while <37% of patients classified as UC1 (Fig. 5f) responded to IFX. Taken together, these results suggest that transcriptional profiles toward UC1 or UC2 are associated with a response to IFX, both in adult and pediatric ulcerative colitis.

## Discussion

A systematic study demonstrated that biopsy sampling was the major source of inter-patient variability[22]. Therefore, such technical variations can mask real biological differences, even though UC is known to present a high level of variability in macroscopic and endoscopic scoring among patients[1,2,8]. To solve this, we limited the analysis to the relevant genes for inflammation, including the phases of tissue repair and regeneration. By using

the key DEGs obtained by a mouse model of colitis, we were able to "ignore" genes that were highly variable between patients (e.g., as a result of technical variation), and focus only on those that contribute to inflammation. This allowed us to temporally assign IBD-risk genes and molecularly subclassify UC patients into two profiles: one of these characterized by genes involved in neutrophil recruitment, activation, and degranulation, and by low response to biologicals.

Different experimental models to study mucosal immune processes associated with the pathogenesis of UC are available[34,35]. Among them, the DSS-induced colitis model is broadly used, due to its simplicity and applicability with different therapeutic drugs[36]. Early studies characterized the temporal changes by qPCR for a handful of inflammatory markers[37], but how noninflammatory (i.e., repair-related) genes fluctuate over time during tissue repair was unknown. Others had previously performed a kinetic microarray analysis only during the acute inflammation phase of DSS (from days 0 to 6)[38], but whether those genes continue to be expressed during tissue repair remained unclear. Moreover, although the DSS-induced colitis model has been extensively used for the study of UC, an open reference for gene expression during intestinal inflammation and tissue repair was still missing. Here, we used a time-series transcriptional characterization of colitis, which allowed us to identify which of the genes contribute to most of the nuances of inflammation over time. In addition, this paper provides an open data source that can be further investigated by others with different questions. As an example, we provided a temporal assignment of IBD-risk genes that might offer insight into their potential functions. Finally, our data show that the DSS mouse model is a relevant model for studying certain aspects of human UC.

Previous studies identified the molecular differences between responder and non-responder IBD patients[13]. These studies were purposely biased by an a priori knowledge of the responder and non-responder IBD samples. In contrast, we successfully classified the patients using a completely unsupervised approach, and therefore, we have potentially identified genes that go beyond the responsiveness to the therapy, by describing the molecular signature of the identified profiles. We were able to do this by using the key DEGs found in the mouse model of colitis, by "debiasing" the human analysis, by "ignoring" genes that were highly variable between patients, and by focusing only on those genes that contribute to inflammation. Consequently, we identified two transcriptomic profiles of ulcerative colitis (UC1 and UC2) based on both adult and pediatric patients.

While per definition, both UC1 and UC2 subpopulations are considered inflamed, only UC1 patients present a higher expression of genes associated with neutrophil degranulation and cytokine signaling, and only 10% of these patients responded to biological therapies. Similar to our results, others have shown that IL-6, IL11, IL13RA, STC1, and PTGS2 were downregulated in patients responsive to IFX[13] (namely UC2 in our study). Another recent report showed that the gene OSM is upregulated in IBD patients compared with healthy controls and is predictive of anti-TNF responsiveness[39]. However, we did not find OSM as differentially expressed between UC1 and UC2 patients. For VDZ, however, a signature for prediction of response to therapy was still missing[15]. While UC1 and UC2 likely define two transient inflammatory states of the colon, the causes of origin and development of such a conserved signature remain unknown.

The identification of UC2, which is characterized by responsiveness to both IFX and VDZ, may have direct implications in the clinical setting. For example, it indicates that UC2 patients would benefit from a treatment with IFX only, since IFX therapy has a higher response rate[6] and is more cost-effective compared with VDZ[40]. On the other hand, identification of non-responsiveness to both IFX and VDZ in the UC1 patient transcriptomic profile, suggests that another line of therapy should be applied. For example, we observed that the B-cell activation factor (TNFSF13B, protein BAFF) was to be found upregulated in UC1 patients. This suggests a potential role of B cells in UC1. Moreover, B cells are known to enhance inflammatory responses by cytokine secretion, such as TNF and IL-6[41], which are also upregulated in UC1 patients. B-cell depletion using an anti-CD20 antibody in a small cohort showed a trend in reducing inflammation, although nonsignificant[42]. However, it remains possible that B-cell depletion might affect only UC1 patients, but not UC2. Similarly, we also observed that UC1 patients have a higher expression of genes involved in the JAK/STAT signaling pathway (PTP4A3, SOCS3) and cytokine signaling (IL6 and IL1B), suggesting a potential role of other therapies for this transcriptomic profile, such as canakinumab (anti-IL-6 mAb), siltuximab (anti-IL-1β mAb), JMS-053 (PTP4A3 inhibitor), and others might apply.

In summary, we have performed an unbiased characterization of the inflammatory and tissue repair processes, using a mouse colitis model, providing a useful resource for understanding colonic inflammation. Many of the genes identified in mice were also detected in human UC patients, thus allowing us to explore the temporal expression of IBD-risk genes during the course of inflammation and gain useful insights into their potential function. Furthermore, they allowed us to identify for the first time two clinically relevant molecular ulcerative colitis profiles (UC1 and UC2) in an unsupervised manner in both adult and pediatric UC patients. Thus, our methodology identified two molecularly distinct UC profiles and serves as a proof of concept for the use of transcriptomic data from highly controlled mice experiments to perform unsupervised and biologically driven analysis of highly variable human data sets.

## Methods

**Mice and induction of DSS colitis.** Animal experiments were done following institutional guidelines of the Stockholm Regional Ethics Committee under approved ethical permit number N89/15. Female 8–12-week–old C57BL/6J mice were obtained from ScanBur (Charles River, Germany) and housed in environmentally enriched ventilated cages under specific pathogen-free conditions (SPF) at the Astrid Fagræus laboratory (AFL, Karolinska Institutet) under 12-h light cycle and receiving water and ration ad libitum (RM1(P), Special Diet Services). For induction of colitis, 2.5% w/v dextran sulfate sodium (DSS; Affymetrics) was supplemented in drinking water and given to mice for 7 consecutive days, with a change on day 3. After the treatment was ceased, mice returned to receive standard water. Mice were monitored everyday for alterations in body weight, disease activity index (DAI)[43]. Mice were anesthetized with isoflurane and killed for blood and tissue sampling.

**Mouse gene expression by mRNA sequencing.** Colon samples were stored in RNAlater (Ambion) at −80 °C until further use. Colonic samples were homogenized using a bead-beating system (Precellys) for total RNA purification, using RNAeasy kit (Qiagen) following the manufacturers' recommendations. RNA purity and quantity were measured by NanoDrop spectrophotometer (ThermoFisher). All samples were screened for RNA integrity check and presented RIN values above 8 on 2100 Bioanalyzer instrument (Agilent). Samples were submitted to Novogene for library preparation, using TruSeq Stranded mRNA Library Prep Kit (poly-A selection) and sequencing on HiSeq-2500 platform (Illumina). Samples were sequenced using single-end 50-bp sequencing[44], aiming a coverage of 20 M reads. Read quality was inspected using MultiQC[45], trimmed with Trimmomatic[46], and further proceeded for abundance estimation using Kallisto[47].

Further data analysis was done in R programming language (Rstudio). Genes with an absolute read count <5 in at least three samples were considered with low expression and filtered out. Differences in tissue cell composition that could affect transcriptional pools were balanced by means of removing unwanted variation, based on negative control genes, using the RUVg function implemented in RUVseq package[48]. Analysis revealed that library sizes strongly correlated with several known intestinal housekeeping genes, such as Hprt ($r = 0.87$) and Gapdh ($r = 0.85$), but not Actb ($r = 0.68$). Moreover, genes such as Cd63 (0.94), Trappc ($r = 0.97$), and Cpped1 (0.97) and Slc25a3 ($r = 0.96$) correlated even more strongly to the library sizes, indicating potentially novel housekeeping genes during colonic inflammation. Negative controls genes were thus defined as genes with a positive Pearson correlation above 0.9 to their respective sample library sizes. Estimated unwanted variation vectors were then used as covariates for calculation of differentially expressed genes (DEGs) using EdgeR package[49]. EdgeR is specialized in performing time-series differential expression by means of generalized linear model (glm) function[25], where time points were parsed as independent factors in the contrast matrix, thus allowing detection of differentially expressed genes at any given time point. Genes were considered differentially expressed when the overall false discovery rate (FDR) < 0.01 and at least one time point had fold change > 1.5. DEGs identified in this manner were used for dimensionality reduction by principal component analysis (PCA), from which genewise contribution to the total variation can be calculated.

Identification of gene modules was done based on smoothed temporal expression curves[50]. Briefly, genewise log-fold changes were smoothed using spline curves and further grouped into modules by using inverse Pearson correlation as the distance for hierarchical agglomerative clustering with Ward's method ("ward. D2"). Functional gene annotation was performed on each gene module individually, using the Gene Ontology (GO_Biological_Process_2017) and the Kyoto Encyclopedia of Genes and Genomes (KEGG_2016) libraries with enrichR package[51].

**UC and IBD-risk gene mapping to the murine RNA-seq data set.** To identify which of the genes are shared between mouse and human ulcerative colitis, we compared the list of DEGs identified by the DSS data set and the list of genes identified by Taman et al.[26]. Mapping of IBD-risk genes was done, using the list of IBD-risk genes identified by fine-mapping at the single-loci resolution[10]. Identification of enriched GO processes and KEGG pathways was done using enrichR[51].

**Classification of adult UC molecular subtypes.** To investigate whether the nuances of inflammation observed in the mouse model could also be found in humans, we made use of four human microarray data sets from GSE12251[13], GSE73661[15], GSE23597[12], and GSE16879[14]. Combined, these data sets contain gene expression and metadata of 447 patients, containing information such as disease type (UC or CD), Mayo macroscopic score, the therapy given, when the sample was collected, and the response to infliximab (IFX) or to vedolizumab (VDZ). Across all data sets, patients were considered inflamed if presenting a Mayo score of 2 or 3 (out of 3). Similarly, patients were considered to respond to therapy, when the respective Mayo score reduced to 0 or 1, between 4 and 8 weeks of treatment with IFX or between 6 and 52 weeks of treatment with VDZ. For this

study, we included only patients with UC before receiving any therapy (either IFX or VDZ), comprising a total transcriptional profiles of 143 patients, of which 102 received IFX and 41 for VDZ.

Probes with $\log_2$ fluorescence count lower than 6 in at least 10 samples were excluded from the analysis. Batches between the data set were observed and corrected using the ComBat function in SVA package[52]. Selection of genes for further exploration was done by different approaches: (1) using all genes; (2) using only the top 100 highly variable genes; (3) using the genes with top 100 high dispersion; (4) the gene with high loading in principal component 1; and (5) the gene with high loading in principal component 2.

We determined whether clustering patterns exist by four independent methods: (1) by dimensionality reduction using t-SNE. Since data originated from biopsies are known to present high variability across patients[22], dimensionality reduction and visualization was done using t-stochastic neighbor embedding (t-SNE). Because of its nonlinear characteristics, t-SNE becomes less sensitive to noise and outperforms PCA[53] to discriminate biopsies based on shared expression patterns, rather than their absolute expression values; (2) by visual assessment of clustering tendency (VAT) using dissimilarity matrices[16]; (3) by using the Hartigan's dip test[19,20], which tests whether the gene distribution is different from a unimodal distribution. Values close to 1 indicate that the data are unlikely to present cluster substructures. We performed bootstrapping 100 times on 90% of the samples to calculate Hartigan's dip test $p$-value. The comparison between bootstrapping with human highly variable genes and mouse PCs (see below) was done using paired Mann–Whitney test; (4) by dividing patients into subgroups using hierarchical agglomerative clustering. Cluster stability was determined by bootstrapping 300 times on 90% of the samples, resulting in the approximate unbiased (AU) statistics[21]. Clusters with AU closer to 100 present higher stability.

Instead of using the top variable genes as above, we alternatively used the top genes identified in the mouse RNA-seq DSS colitis data set (see above). To this end, the top 100 genes identified in PC1 and PC2 were selected for identification of the respective human homologs. Together, 175 genes were found in top genes in both PC1 and PC2 and from these, 148 genes had a homolog in humans. In total, 57 homolog genes were found between our mouse PCs and the human data set. Dimensionality reduction was performed with t-SNE. Assessment of the clustering tendency was done as described above. Agglomerative clustering on the Euclidean distance using complete linkage was used to discriminate patient subgroups UC1 and UC2. For the matter of definition in this study, patients that present a higher mean expression of the 57 mouse–human homologs were classified as UC1, while those with low expression were classified as UC2. Differences in expression between UC1 and UC2 were calculated, using eBayes method in limma package[54]. Probes with fold changes above 1.5 and FDR lower than 0.001 were considered significantly differentially expressed. Identification of enriched GO, KEGG, and cell types was done using enrichR[51].

To identify which of the genes can discern UC1 from UC2, we trained a logistic regression classifier for each gene individually and compared it with the UC1 and UC2 classification mentioned above. The sensibility and sensitivity of the prediction were summarized using the area under the curve (AUC) method. Genes with AUC values closer to 1 (100%) have a better accuracy to distinguish UC1 and UC2 patients.

**Classification of UC molecular subtypes in pediatric patients**. In addition to using mouse genes to stratify adult UC patients (see above), we applied a similar strategy to a RNA-seq data set from pediatric UC patients[33]. This data set contains the expression levels and detailed metadata information of 206 colonic samples. After failing to respond to first-line therapy, all patients in this cohort received infliximab and the response was evaluated after 4 weeks. Genes with read count <5 in at least 10% of the samples were considered with low expression and filtered out. Batches associated with sex chromosomes were detected and corrected using ComBat[52]. Counts were normalized by TMM normalization method implemented in EdgeR package[49], and subsequently used for stratification, using the genes in PC1 and PC2 identified in the mouse model of colitis (see above). EdgeR and limma packages estimate differential expression by different assumptions, and therefore result in slightly different results[55]. Thus, to allow fair comparison between the results found between the microarray data set and the RNA-seq, we opted to use the same differential expression strategy in both data sets. Differences in expression between UC1 and UC2 were calculated using eBayes method in limma package[54], using $\log_2$-transformed counts per million (instead of raw counts). Strict cutoffs were used to ensure result robustness that also accounts for the differences in sample size and methodologies between the data sets. Genes with fold changes above 1.5 and FDR lower than $1^{-10}$ were considered significantly differentially expressed. Comparison between both data sets was done using Venn diagrams, and by comparing FDR statistics and $\log_2$ fold changes in gene expression. Finally, differences in metadata parameters between UC1 and UC2 were evaluated using chi-square or with Mann–Whitney tests when applicable.

**Lamina propria cell isolation for analysis by flow cytometry**. Cell isolation from the colonic tissue was performed as previously described[56] with modifications.

Briefly, tissues were open longitudinally, cut into 1-cm pieces, and washed with PBS. The epithelial cell fraction was obtained by incubating the tissue with Buffer-A (PBS, 5% FCS, and 5 mM EDTA) at 37 ˚C for 20 min under agitation at 600 rpm. The supernatant was collected and kept on ice, while the remaining tissue was washed two times with PBS. Tissue was digested with collagenase solution containing 0.15 mg/ml Liberase TL (Roche) and 0.1 mg/ml DNase I (Roche) in HBSS and incubated at 37 ˚C for 60 min under agitation at 1200 rpm. The digested and the epithelial cell fraction were mixed, filtered through a 100-μm cell strainer, pelleted by centrifugation at 1750 rpm, and resuspended in Buffer-A. Cell suspensions were blocked with Fc-blocking solution (1:1000, eBioscience) and stained with the antibody mix (1:200), both at 4 ˚C for 15 min. The following antibodies were purchased from BD Biosciences: CD45.2 (104), CD3 (500A2), CD90.2 (53-2.1), EPCAM (G8.8), CD11b (M1/70), CD11c (N418), Ly6G (1A8), B220 (RA3–6B2), and CD64 (54-5/7.1). The following antibodies were purchased from eBiosciences: CD103 (2E7) and Ly6C (HK1.4). Counting beads (Spherotech) and DAPI (1:400, Sigma) were added to each sample to allow absolute cell quantification and exclusion of dead cells. Data acquisition was done using 5-laser LSR Fortessa flow cytometer (BD Biosciences) and analysis was carried out with FlowJo software (TreeStar).

**Histological analyses**. The colonic tissue was rinsed and flushed with PBS, and gently squeezed out to remove non-adherent bacteria, fixed in 4% formaldehyde solution for 24 h, and embedded in paraffin. Five- micrometer sections were stained with H&E. Ki67 (1:100, Cat# MA5-14520, Thermo Scientific) staining was performed according to a previously published protocol[57]. A pathologist accessed the tissue pathological score in a blind manner and scored the sections as previously described[58].

**FITC-dextran assay**. Assessment of the epithelial barrier integrity by FITC dextran was done as previously described[59]. Mice were gavaged with 10 mg/mL FITC dextran (Sigma) at different time points of DSS colitis (see above), but on the same day of killing. Four hours later, mice were killed and the blood was collected for analysis. Sera were diluted 1:1 v/v in PBS and added to a 96-well plate for fluorescent-based assays (Invitrogen), and were quantified on a fluorescent plate reader using a 535/587-nm ex/em filter. FITC-dextran concentration was calculated by interpolation to 12-dilution FITC-dextran standard curve.

**Statistical analyses**. Statistical analyses were performed using Prism Software 6.0 (GraphPad). Two-sample comparisons were compared using a two-tailed Student's $t$ test. ANOVA with Dunnett's post hoc was used for calculation of significance at multiple time points relative to the control (day 0). Noncontinuous data were compared using a nonparametric Mann–Whitney $U$ test. The results were considered significant when $p < 0.05$.

**Reporting summary**. Further information on research design is available in the Nature Research Reporting Summary linked to this article.

## Data availability

All the raw data generated in this study were deposited at the Gene Expression Omnibus under assession number GSE131032.

## Code availability

Codes used in this paper are available on Github (https://github.com/czarnewski/uc_classification).

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

## Acknowledgements

We would like to thank Stefan Bonn, Samuel Huber, and Charlotte Hedin for critical reading and suggestions on the paper. We thank Elaine Hussey for editorial assistance. EJV was supported by grants from the Swedish Research Council VR grant K2015-68X-22765-01-6, Formas grant nr. FR-2016/0005, and Wallenberg Academy Fellow (WAF) program.

## Author contributions

P.C. and E.J.V. conceived the idea and wrote the paper. P.C. performed bioinformatics analysis and created schematic illustrations. N.G. and E.J.V. provided reagents and guidance. P.C., S.M.P., S.D., C.S. and O.E.D. performed the experiments. P.C., N.G. and E.J.V. analyzed and interpreted the data. All authors contributed to paper writing.

## Additional information

**Competing interests:** The authors declare no competing interests.

