## [Peer Review File · Nature Communications]

Reviewers' comments:

Reviewer #2 (Remarks to the Author):

Major Findings: The authors have conducted an innovative forward translation approach, using longitudinal colon gene expression analysis from the acute epithelial injury and recovery phases of DSS colitis in mice, to inform molecular classification of Ulcerative Colitis (UC) patients. Using publically available microarray data from trials of two biologic therapies in adult UC, they show that patient classification at the colon transcription level is highly variable and unstable. A careful time series of colon inflammation and repair following DSS administration to mice revealed specific inflammatory, epithelial repair, and metabolic modules during the different stages of disease in mice, some of which were conserved in humans. They then mapped these temporal data from the murine model to human risk genes, providing potential insight into their role in UC pathogenesis. Finally, in the most important finding of the study, they show that conserved inflammatory genes largely involved in neutrophil function (although likely to some extent also monocyte/macrophage function), now provided stable classification of two sub-types of UC in humans. The response to either anti-TNF or anti-integrin biologic therapy was significantly worse in patients expressing the neutrophil inflammatory gene module.

Approach & Conclusions: The analytic approaches are all quite appropriate, and the forward translation from the murine model to humans is novel and will be of interest to the field. A fundamental problem is whether current murine models reflect at least some aspects of human UC pathogenesis, and this report helps to address that. The results are very clearly presented, and all strongly support the authors' conclusions. Certainly this work could be reproduced from the information provided.

Further Evidence: Patient classification is based on four publically available cohorts including 102 adult UC patients. Ideally the authors would demonstrate replication of the classifier on an independent cohort of similar size. Unfortunately the initial innate gene expression classifier developed from one of the microarray data sets used here has not subsequently replicated in further clinical trials, containing some of the same innate genes.

Novelty: With the exception of the forward translation analytic approach from the murine data, the main results in Figs. 4 & 5 essentially validate and replicate what has already been reported by West et al (Nat Med. 2017 May; 23(5):579-589), which in addition showed stromal cell derived OSM as a key driver of this inflammatory pathway, and Haberman et al (Nat Commun. 2019 Jan 3; 10(1):38), which utilized a larger cohort of treatment naïve pediatric UC patients, and provided potential links of a similar innate inflammatory module to both microbiota and treatment response. Therefore, while the approach linking the murine to human data, and validation of this non-response gene signature, will be of interest, it may not at this point be viewed as a completely new finding.

Reviewer #3 (Remarks to the Author):

This study by Czarnewski et al (Villablanca group) explores expression analysis in colonic samples from previously published patients with UC. Unable to distinguish UC subgroups using an unsupervised classification of prior transcriptomic data of biopsy specimens, the authors perform a kinetic transcriptomic study of the DSS model of colitis. This demonstrated gene groups differentially regulated during inflammatory and recovery disease phases. Applying this smaller data set and its even smaller overlap onto the human UC specimens, they are then able to subcategorize the UC patients into 2 groups. Remarkably, the authors demonstrate that these groups, from previously published studies, demonstrate markedly different responses to immune therapies (infliximab and vedolizumab).

This last observation on differential responsiveness of the UC1 and UC2 subgroup demonstrate that

the team's descriptive studies of gene expression have led to meaningful insight into the use of gene expression to classify responders and non-responders to therapy, and indicates the potential of their approach.

Major comments:

1. This is a well performed and clearly described study yielding an interesting finding on the use of limited transcriptomics to classify responder and non-responders to therapy that is of interest and relevance to the field.
2. The authors in describing their UC1 and UC2 subgroups imply that these are distinct UC populations, stating that patients themselves are classified into subgroups. The authors should be clearer to the fact that they are only classifying biopsy specimens into the 2 subgroups; their data does NOT indicate that these are in fact true patient subclasses with different physiologies. It may for instance be that these patient groups merely are at different phases within their disease and that impacts response to therapy or there may be other overriding differences between the UC1 and UC2 subgroups that have led to the simultaneous correlative differences in outcome and transcriptomics.
3. Although the authors describe their results as an unsupervised molecular stratification of UC patients, it is necessary to point out that the data is purely retrospective and without separate validation cohorts. Although the authors use a vedolizumab cohort and infliximab cohorts (combined into a single cohort), their analyses lack separate verification data sets (or better) prospective confirmation.
4. The clinical response data indicates that the methodology has biological significance and is fundamental to the manuscript. Yet, this data segregates patients into simply responder or non-responders. How were these classifications made? Is it possible to quantify response rather than merely digitally classify into response categories? Is there additional clinical data that is available from the studies for which this information was extracted to provide a better understanding of the clinical differences between the UC1 and UC2 groups (e.g. patient attributes that may correlate with response; demonstrating no difference in fundamental demographic attributes of the patient populations would be helpful to supporting the case that these truly are different populations)? It would have been hoped that a more complete multiparametric analysis be performed.

Minor comments:

1. Please label the supplemental tables. These are not individually labeled and linked with the manuscript in the files received.

Response to Reviewer's comments (NCOMMS-19-00868-T):

We are grateful for the reviewer's feedback with regard to improving our manuscript and appreciate the suggestions and guidance. Please find below a point-by-point response to the reviewer's concerns. Changes to the revised manuscript are highlighted in yellow.

Reviewer #2 (Remarks to the Author):

Major Findings: The authors have conducted an innovative forward translation approach, using longitudinal colon gene expression analysis from the acute epithelial injury and recovery phases of DSS colitis in mice, to inform molecular classification of Ulcerative Colitis (UC) patients. Using publically available microarray data from trials of two biologic therapies in adult UC, they show that patient classification at the colon transcription level is highly variable and unstable. A careful time series of colon inflammation and repair following DSS administration to mice revealed specific inflammatory, epithelial repair, and metabolic modules during the different stages of disease in mice, some of which were conserved in humans. They then mapped these temporal data from the murine model to human risk genes, providing potential insight into their role in UC pathogenesis. Finally, in the most important finding of the study, they show that conserved inflammatory genes largely involved in neutrophil function (although likely to some extent also monocyte/macrophage function), now provided stable classification of two sub-types of UC in humans. The response to either anti-TNF or anti-integrin biologic therapy was significantly worse in patients expressing the neutrophil inflammatory gene module.

We thank Reviewer #2 for reading our manuscript carefully and for identifying the major findings it provides.

Approach & Conclusions: The analytic approaches are all quite appropriate, and the forward translation from the murine model to humans is novel and will be of interest to the field. A fundamental problem is whether current murine models reflect at least some aspects of human UC pathogenesis, and this report helps to address that. The results are very clearly presented, and all strongly support the authors' conclusions. Certainly this work could be reproduced from the information provided.

We thank Reviewer #2 for these comments and in particular, for recognizing the novelty and reproducibility of our work.

Further Evidence: Patient classification is based on four publicly available cohorts including 102 adult UC patients. Ideally the authors would demonstrate replication of the classifier on an independent cohort of similar size. Unfortunately the initial innate gene expression classifier developed from one of the microarray data sets used here has not subsequently replicated in further clinical trials, containing some of the same innate genes.

We agree with Reviewer #2 regarding the use of a cohort to validate our findings. We therefore performed a new set of experiments on an independent cohort. We applied the analysis strategy to the new publicly available dataset from Haberman et al (Nat Commun. 2019 Jan 3;10(1):38). This dataset contains RNA-seq data of 206 pediatric UC patients prior to IFX therapy. We performed analysis of this dataset following the same strategy used in the microarray dataset, and observed that patients also segregated into 2 transcriptomic profiles. In fact, we also performed an in-depth stringent comparison of the microarray (adult) and RNA-seq (pediatric) datasets, showing that over 100 genes were conserved between the analysis. This is highly significant to distinguish UC1

from UC2. This is discussed on pages 15-16 (lines 288 to 314) and on page 18 (lines 351-352) with Fig 5d-f and two additional supplementary figures (S8 and S9).

Novelty: With the exception of the forward translation analytic approach from the murine data, the main results in Figs. 4 & 5 essentially validate and replicate what has already been reported by West et al (Nat Med. 2017 May;23(5):579-589), which in addition showed stromal cell derived OSM as a key driver of this inflammatory pathway, and Haberman et al (Nat Commun. 2019 Jan 3;10(1):38), which utilized a larger cohort of treatment naïve pediatric UC patients, and provided potential links of a similar innate inflammatory module to both microbiota and treatment response. Therefore, while the approach linking the murine to human data, and validation of this non-response gene signature, will be of interest, it may not at this point be viewed as a completely new finding.

We agree with Reviewer #2 that our analysis validates and replicates previous reports. However, unlike these reports, we performed an unsupervised molecular stratification that leads to two distinct groups in which one (UC2) was more likely to respond to biological treatments while the other (UC1) was not. Therefore, we believe that validating published data is actually a strength of our manuscript.

We would like to further explain this by pointing out that West et al 2017 and Haberman et al 2019 used a retrospective strategy. Patients were first segregated into responders (R) and non-responders (NR) for a certain biological treatment followed by an analysis of differentially expressed genes (before therapy). This approach has some limitations; the list of genes identified is therapy specific. In other words, the results obtained for a hypothetical therapy A were only valid for that therapy, and another cohort needs to be evaluated to check the signature for response to a different therapy. Moreover, the analysis is restricted to the dataset used (also known as overfitting). In our manuscript, we apply patient stratification in a prospective manner, and try to overcome the limitations above. Herein, patients were first subdivided based on their transcriptional signature (assisted by the mouse signature) and only then assessed for response to biologicals. The implications are:

- 1) The patient stratification is purely based on a biological transcriptomic signature without any a priori knowledge on their response to biologicals, that is, unsupervised. In other words, our classification can be applied to any patient, regardless of the therapeutics he/she receives. Therefore, even though we performed the analysis using infliximab and vedolizumab datasets, the UC1 / UC2 stratification is applicable to any other therapy (i.e. corticosteroids).
- 2) Biologically, instead of dividing patients by response to therapeutics, we looked for intrinsic differences in patient transcriptional signatures, which in the end turned out to match well with the response to both infliximab and vedolizumab. This makes our findings more robust and biologically meaningful.

Overall, our approach/method is unsupervised and it stratifies patients based on their intrinsic inflammatory signature, and not on their response to therapy. Also, validation of previous data indicates the validity of our method.

Reviewer #3 (Remarks to the Author):

This study by Czarzewski et al (Villablanca group) explores expression analysis in colonic samples from previously published patients with UC. Unable to distinguish UC subgroups using an unsupervised classification of prior transcriptomic data of biopsy specimens, the authors perform a kinetic transcriptomic study of the DSS model of colitis. This demonstrated gene groups differentially regulated during inflammatory and recovery disease phases. Applying this smaller data set and its even smaller overlap onto the human UC specimens, they are then able to subcategorize the UC patients into 2 groups. Remarkably, the authors demonstrate that these groups, from previously published studies, demonstrate markedly different responses to immune therapies (infliximab and

vedolizumab).

This last observation on differential responsiveness of the UC1 and UC2 subgroup demonstrate that the team's descriptive studies of gene expression have led to meaningful insight into the use of gene expression to classify responders and non-responders to therapy, and indicates the potential of their approach.

We thank Reviewer #3 for carefully evaluating our manuscript, for the helpful comments and for the interest expressed in our work.

Major comments:

1. This is a well performed and clearly described study yielding an interesting finding on the use of limited transcriptomics to classify responder and non-responders to therapy that is of interest and relevance to the field.

We thank Reviewer #3 for appreciating the study.

2. The authors in describing their UC1 and UC2 subgroups imply that these are distinct UC populations, stating that patients themselves are classified into subgroups. The authors should be clearer to the fact that they are only classifying biopsy specimens into the 2 subgroups; their data does NOT indicate that these are in fact true patient subclasses with different physiologies. It may for instance be that these patient groups merely are at different phases within their disease and that impacts response to therapy or there may be other overriding differences between the UC1 and UC2 subgroups that have led to the simultaneous correlative differences in outcome and transcriptomics.

We agree with Reviewer #3. The UC1 and UC2 transcriptomic profiles very likely indicate a transient molecular inflammatory signature rather than a genetic-dependent response of patients. This is also in line with the time-wise inflammatory activity observed in mice which presents distinct inflammatory profiles whilst having the identical genetic background. Therefore, UC2 patients may ultimately 'become' UC1 if the inflammation persists for a long time, or *vice versa*. In fact, a follow-up study is necessary to further verify this hypothesis. We thank Reviewer #3 for pointing this out and we made changes throughout the whole manuscript, replacing terms such as "UC1/UC2 subtype / subgroup" to "UC1/UC2 transcriptomic profile" where appropriate. To explicitly clarify this concept, we also added to the Discussion, the following sentence: "Whilst UC1 and UC2 likely define two transient inflammatory states in the colon, the causes of origin and development of such a conserved signature remain unknown." (lines 361 to 362).

3. Although the authors describe their results as an unsupervised molecular stratification of UC patients, it is necessary to point out that the data is purely retrospective and without separate validation cohorts. Although the authors use a vedolizumab cohort and infliximab cohorts (combined into a single cohort), their analyses lack separate verification data sets (or better) prospective confirmation.

We agree with Reviewer #3 regarding the necessity of an independent dataset to corroborate our findings. In contrast to recent papers regarding response to therapy (see below), our unsupervised strategy is essentially a prospective approach. Therefore, we would like to provide a more detailed explanation of the differences and such consequences.

West et al 2017 and Haberman et al 2019 applied a retrospective strategy. Patients were first segregated into responders (R) and non-responders (NR) for a certain biological treatment. After that, differentially expressed genes (before therapy) were analyzed. This approach has some limitations, for example, the list of genes identified is therapy-specific. In other words, the results obtained for a hypothetical therapy A are only valid for

that therapy, and another cohort needs to be evaluated to check the signature for response to a different therapy. Moreover, the analysis is restricted to the dataset used (also known as overfitting). In our manuscript, we apply patient stratification in a prospective manner, and try to overcome the limitations above. Patients were first subdivided based on their transcriptional signature (assisted by the mouse signature) and only then assessed for response to biologicals. The implications are:

- 3) The patient stratification is purely based on biological transcriptomic signature without any a priori knowledge on their response to biologicals (that is, unsupervised). In other words, our classification can be applied to any patient, regardless of the therapeutics he/she receives. Therefore, even though we performed the analysis using infliximab and vedolizumab datasets, the UC1 / UC2 stratification is applicable to any other therapy (i.e. corticosteroids).
- 4) Biologically, instead of dividing patients by response to therapeutics, we looked for intrinsic differences in patient transcriptional signatures, which in the end turned out to match well with the response to both infliximab and vedolizumab. This makes our findings more robust and biologically meaningful.

Overall, our approach/method is unsupervised and it stratifies patients based on their intrinsic inflammatory signature, and not on their response to therapy. The validation of previous data indicates the validity of our method.

To complement our analysis, we applied the analysis strategy to a new publicly available dataset from Haberman et al (Nat Commun. 2019 Jan 3;10(1):38). This dataset contains RNA-seq data of 206 pediatric UC patients prior to IFX therapy. We analyzed this dataset following the same strategy used in the microarray dataset, and observed that patients also segregated into 2 profiles, sharing many genes with our previous analysis. In fact, we also performed an in-depth stringent comparison of the microarray (adult) and RNA-seq (pediatric) datasets, showing that over 100 genes were conserved between the analysis, which are highly significant to distinguish UC1 from UC2. This is discussed on pages 15-16 (lines 288 to 314) and page 18 (lines 351- 352), and shown in Fig 5d-f and in two additional supplementary figures (S8 and S9).

4. The clinical response data indicates that the methodology has biological significance and is fundamental to the manuscript. Yet, this data segregates patients into simply responder or non-responders. How were these classifications made? Is it possible to quantify response rather than merely digitally classify into response categories? Is there additional clinical data that is available from the studies for which this information was extracted to provide a better understanding of the clinical differences between the UC1 and UC2 groups (e.g. patient attributes that may correlate with response; demonstrating no difference in fundamental demographic attributes of the patient populations would be helpful to supporting the case that these truly are different populations)? It would have been hoped that a more complete multiparametric analysis be performed.

Reviewer #3 has a valid point that better patient demographics would be helpful to further translate our findings to a clinical setting. Since we made use of 4 publicly available datasets, the detailed information regarding gender, ethnicity, age, clinical records are not accessible to us in the microarray dataset. However, we recently gained access to and performed an additional analysis on the 206-patient pediatric UC cohort recently published by Haberman et al (Nat Commun. 2019 Jan 3;10(1):38) (New Fig.5)), which indeed contained a more well annotated dataset with additional metadata of interest. We found that age of diagnostics, sex, histological score and calprotectin levels were not associated with UC1 and UC2 profiles. In addition, Total Mayo and Pucal scores did indeed associate significantly with our unbiased biologically-driven UC1/UC2 profiles. These results were added on pages 15-16 in the main text, with supplementary figures S8 and S9.

Minor comments:

1. Please label the supplemental tables. These are not individually labeled and linked with the manuscript in the files received.

We thank Reviewer #3 for pointing this out and apologize for the inconvenience. We made changes to all the supplemental tables, including the table legend and labels inside each file.

REVIEWERS' COMMENTS:

Reviewer #2 (Remarks to the Author):

The authors have carefully considered the comments and provide new analyses further supporting their main conclusions. No further comments.

Reviewer #3 (Remarks to the Author):

The authors have done a good job addressing all concerns raised.